# Applicability of Electron-Beam and Hybrid Plasmas for Polyethylene Terephthalate Processing to Obtain Hydrophilic and Biocompatible Surfaces

**DOI:** 10.3390/polym16020172

**Published:** 2024-01-06

**Authors:** Tatiana Vasilieva, Elena Nikolskaya, Michael Vasiliev, Mariia Mollaeva, Margarita Chirkina, Maria Sokol, Nikita Yabbarov, Tatiana Shikova, Artem Abramov, Aleksandr Ugryumov

**Affiliations:** 1Joint Institute for High Temperatures of Russian Academy of Sciences, Izhorskaya st. 13 Bd. 2, 125412 Moscow, Russia; vasilev.mn@mipt.ru; 2Emanuel Institute of Biochemical Physics, Russian Academy of Sciences, Kosygina st. 4, 119334 Moscow, Russia; elenanikolskaja@gmail.com (E.N.); mollaevamariia@gmail.com (M.M.); chir.marg@mail.ru (M.C.); mariyabsokol@gmail.com (M.S.); yabbarovng@gmail.com (N.Y.); 3Department of Electronic Devices and Materials, Ivanovo State University of Chemistry and Technology, Sheremetevskiy Prospect 7, 153000 Ivanovo, Russia; shikova@isuct.ru; 4TVEL JSC, Kashirskoye Shosse 49, 115409 Moscow, Russia; argabramov@tvel.ru (A.A.); avugryumov@tvel.ru (A.U.)

**Keywords:** polyethylene terephthalate, plasma chemical polymers modification, biocompatibility, non-temperature plasma, RF discharge, electron-beam plasma, hybrid plasma

## Abstract

The applicability of beam-plasma chemical reactors generating cold hybrid plasma for the production of noncytotoxic polymeric surfaces with high hydrophilicity and good biocompatibility with human fibroblast culture and human red blood cells was studied. Oxygen hybrid plasma was excited by the joint action of a continuous scanning electron beam and a capacity-coupled RF-gas discharge. Experiments showed that hybrid plasma treatment caused polar oxygen-containing functional group formation in the surface layer of poly (ethylene terephthalate) films. No thermal or radiative damage in tested polymer samples was found. The plasma-modified polymers turned out to be noncytotoxic and revealed good biocompatibility with human fibroblasts BJ-5ta as well as lower hemolytic activity than untreated poly (ethylene terephthalate). Experiments also demonstrated that no phenomena caused by the electrostatic charging of polymers occur in hybrid plasma because the electron beam component of hybrid plasma eliminates the item charge when it is treated. The electron beam can effectively control the reaction volume geometry as well as the fluxes of active plasma particles falling on the item surface. This provides new approaches to the production of abruptly structured patterns or smooth gradients of functionalities on a plane and 3D polymeric items of complicated geometry.

## 1. Introduction

Over the past decade, organic polymers have gained great importance as advanced materials for biochemical laboratory techniques, medical diagnostics, drug delivery systems, biotechnological procedures, and the fabrication of medical implants [1,2]. So considerable popularity of polymers is due to their remarkable physicochemical parameters, namely low density, high elasticity and specific strength, corrosion resistance, long-term durability, resistance against chemicals and/or abrasion etc. [1,2,3,4]. For example, nowadays polyethylene terephthalate (PET) and its derivatives are the most commonly used fibres in commercial vascular prostheses [5]. Other PET applications are sutures, heart valves, surgical meshes, scaffolds, urinary and bloodstream catheters [6].

However, despite the valuable bulk properties mentioned above, PET has low surface free energy, leading to poor wettability and poor adhesion in contact with living cells, which can cause damage to human tissues [5,6]. Therefore, to improve the hydrophilic properties of the polymeric surface and to increase its biocompatibility modification and additional functionalization are often required.

Technologies based on non-thermal plasmas (NTPs) have many applications, including electrical device microfabrication, biomedicine, dentistry, agriculture, ozone generation, chemical synthesis, materials engineering, coating, disease therapy, and the degradation of hazardous pollutants in the environment. A detailed and comprehensive review of the recent advances in NTPs can be found in [7]. NTPs can be operated at or around body temperature (not higher than 40 °C) and are termed cold plasma [8]. In most cases, cold plasmas are non-equilibrium because temperatures of plasma electrons are much higher than those of ions and neutral particles.

Biomedical applications of NTPs especially of cold plasmas have been actively studied since the end of the first decade of the 2000s. Nowadays, the field of plasma medicine includes several applications of NTPs in biology and medicine: sterilization, disinfection, and decontamination, plasma-aided wound healing, plasma dentistry, cancer therapy (so-called “plasma oncology”), plasma pharmacology, and plasma treatment of implants for biocompatibility improvement [8,9].

Polymeric materials surface modification low-pressure and atmospheric pressure NTPs are frequently considered [10,11,12], since they offer several major advantages over conventional methods like mechanical abrasion, wet chemical cleaning, etching, etc. By hydrophilic or hydrophobic groups insert the plasma-stimulated surface activation changes the surface energy and the water wettability of polymers, affecting their adhesive interface bonding [12,13], specific active molecules binding [14], integration with living tissues [10,11,15] and antibacterial properties [9,16].

For instance, the positive effects of polymeric surfaces NTP treatment were demonstrated in culturing human smooth muscle cells, bone marrow stromal cells, hepatocytes, fibroblasts and many other cell types [15,17]. Thus, plasma-modified hydrophilic polymers are a perspective for tissue engineering as well as clinical applications and optimization of implant surfaces [18].

NTP-processing of organic polymer has been the most thoroughly studied for corona, radiofrequency (RF), microwave, and low-pressure direct-current discharge effects [15,19,20,21,22]. Electron-beam plasma (EBP) reactors, generating large volumes of cold plasma by the injection of an electron beam (EB) into a gaseous medium could also be a promising alternative to gas discharge [23,24]. Under typical conditions (EB power *N_b_* < 1 kW, fore vacuum gas pressure 1 < *P_m_* < 10 kPa and moderate electron energies *E_b_* < 100 keV) the EBP contains chemically active particles in super-equilibrium concentrations even at low, e.g., room, temperatures. Due to this fact, the EBP-reactors are extremely effective for modification of high molecular weight organic substances destructible when overheated [23,24]. Besides, the polymer EBP treatment is a dry, clean, and very fast process requiring minimal chemical consumption.

When the EB and RF discharge interact simultaneously with a plasma-generating medium, the so-called hybrid plasma (HP) is produced [23,24]. This combination of the EBP and RF discharge can be a perspective for polymer functionalization. The main HP-reactors advantages are as follows:The reaction volume is uniform and doesn’t contract with an increase of the plasma generating gas pressure to values at which the RF discharge is filamentary or does not glow at all;The EB attracts RF discharge and controls its location at the surface of the polymeric substrate as well as fluxes of neutral active particles that are produced by RF discharge and are mainly responsible for the wettability increase and the improvement of bioactivity of polymers.

The combination of EBs with microwave plasma has been used for gas cleaning [25] and for the volatile organic compound decomposition [26]. However, plasma–chemical processing of polymeric materials, using the HP is not well studied though the HP has extremely high chemical activity even at room temperatures due to its unique composition (e.g., molecules, atoms, radicals, and ions in stable and excited states, plasma electrons and electrons of the injected beam) and properties.

The present study aimed to compare the poly (ethylene terephthalate) films surface morphology, chemical composition, and wettability as well as their biocompatibility before and after treatment by cold plasma of three types, namely RF discharge, the EBP, and hybrid plasma.

To characterize changes in PET properties due to plasma chemical modification, several analytical techniques were applied [27]. The chemical composition of modified PET was characterized by spectroscopic techniques commonly used in polymer analysis, namely the Fourier transform infrared spectroscopy (FTIR) and X-ray photoelectron spectroscopy (XPS). Since the morphology of polymeric surfaces has a strong impact on their functional properties, it was analyzed by atomic force microscopy (AFM). The acquired wettability of PET films was evaluated by the sessile drop method. Moreover, the bioproperties, i.e., cytotoxicity, protein adsorption, and hemocompatibility of plasma-modified PET were studied.

## 2. Materials and Methods

### 2.1. Materials

#### 2.1.1. Polymeric Materials 

All experiments were carried out with plane PET samples (films produced by the “Polimer-Teknika” company, Vladimir, Russia) 5 × 5 mm in size and thickness of 70 µm. Before plasma chemical treatment the samples were successively cleaned with deionized water (three times for 10 min) in an ultrasonic bath.

#### 2.1.2. Cell Culture Materials 

Phosphate buffered saline (PBS) (Biolot, St. Petersburg, Russia); 3-(4,5-dimethyl-2-thiazolyl)-2,5-diphenyl-2H-tetrasolium bromide (MTT) and Mowiol (Sigma-Aldrich, St. Louis, MO, USA); 96% ethanol (Chimmed, Moscow, Russia); Dulbecco’s modified Eagle’s medium (DMEM) (Gibco, Waltham, MA, USA); fetal bovine serum (FBS) (Gibco, Waltham, MA, USA); 0.9% saline (PanEco, Moscow, Russia); dimethyl sulfoxide (DMSO) (Amreso, Solon, OH, USA); 0.02% EDTA; 0.05% trypsin solutions (Gibco, Waltham, MA, USA); and gentamycin (PanEco, Moscow, Russia) were used for the experiments with cell cultures. BCA protein assay kit, lysozyme (Thermo Fisher, Waltham, MA, USA), BSA, and bovine fibrinogen (type I-S) (Sigma-Aldrich, Burlington, MA, USA). Blood collection tubes (Vacuette, Greiner Bio-One, Kremsmünster, Austria).

### 2.2. Characterization of Plasma-Modified PET Surface

#### 2.2.1. Atomic Force Microscopy

The atomic force microscopy (AFM) surface study was performed with the Veeco Dimension V apparatus (Veeco Instruments Inc., Plainview, NY, USA) in contact mode. Scanning was carried out at a frequency of 0.8–1.01 Hz, with a resolution of 256 × 256 and 512 × 512 points. The obtained images were processed by the NanoScope Software, v1.40r1.

#### 2.2.2. Fourier Transform Infrared (FTIR) Spectroscopy

The PET samples were analyzed by the Fourier transform IR attenuated total internal reflection (ATR) spectroscopy using a Nicolet Avatar 360 spectrometer (Thermo Fisher Scientific, Waltham, MA, USA). The ATR element was a zinc selenide crystal with a 12-fold reflection and a reflection angle of 45°. The signal accumulation mode was experimentally selected based on the results of 32 scans. The resolution was 2 cm^−1^. The spectra were recorded in a wavenumber (*ν*) range of 1400–4000 cm^−1^ and interpreted in accordance with [28,29,30,31].

#### 2.2.3. X-ray Photoelectron Spectroscopy

The chemical composition of polymer surfaces was studied by X-ray photoelectron spectroscopy (XPS) using a Thermo Scientific Theta Probe spectrometer (Thermo Fisher Scientific, USA) with monochromatic AlKα radiation (1486.6 eV). The spectrometer energy scale was calibrated at the Au4f7/2 line with a binding energy of 84.0 eV. The XPS spectra were interpreted in accordance with [32,33,34]. The XPS data were processed using the ThermoAvantage software, version 5 (Thermo Fisher Scientific, Waltham, MA, USA) by Gaussian–Lorentzian functions after subtraction of a Shirley background. The Gaussian full width at half maximum (FWHM) of 1.5 eV was established before peak fitting.

#### 2.2.4. Contact Angle and Surface Free Energy (SFE) Measurements 

The contact angles with deionized water (*θ_W_*) or diiodomethane (*θ_DM_*) (Sigma-Aldrich, Schnelldorf, Germany) were measured by the sessile drop method with the optical contact angle-measurer CAM 101 (KSV Instruments, Espoo, Finland). All measurements were carried out immediately after plasma modification at room temperature and fixed at 45% relative humidity. The SFE (*γ_tot_*) and its polar (*γ_pol_*, deionized water) and dispersive (*γ_disp_*, diiodomethane) components were calculated by the Owens-Wendt method [35].

### 2.3. Cell Culture and Cytotoxic Activity Analysis

The immortalized human fibroblasts BJ-5ta cell line was maintained in 25 cm^2^ polystyrene flasks in the DMEM medium supplemented with 10% FBS and gentamycin (50 µg/mL) at 37 °C in a humidified atmosphere containing 5% CO_2_. The cells were replated using a trypsin-EDTA solution twice per week. 

To assess the cytotoxic activity and biocompatibility, the cells were seeded into 24-well plates (Corning TCT surface treatment, Lowell, MA, USA) (10,000 cells per well in 1 mL of DMEM) onto the treated side of PET film (10 mm in diameter) samples and incubated under standard conditions for 96 h. Cell photos were taken at 24, 48, 72 and 96 h of incubation by a Nikon Diaphot phase contrast microscope at 40× magnification equipped with a Levenhuk M1400Plus camera (Levenhuk, Saint Petersburg, Russia). The standard MTT assay was applied in order to evaluate cell survival [36]. Cells were seeded onto the samples within 2 h after the treatment. Blank culture and sterile non-modified PET were the negative and positive controls, respectively. 4 h before the end of incubation, each well was supplemented with 250 μL of MTT solution (1 mg/mL) in the serum-free DMEM and incubated for 4 h. Next, the medium was aspirated, and precipitated formazan crystals in each well were dissolved in 500 μL of DMSO. Then 100 μL of supernatant from each well was transferred for analysis into a 96-well plate and the absorbance was measured at wavelength λ = 540 nm with a plate photometer (BioRad, Hercules, CA, USA). The cell survival was calculated as (Equation (1)):Cell survival = (*A_experimental_*/*A_control_*) × 100%(1)
where *A_experimental_* is the optical density of supernatant obtained from fibroblasts incubated with original or plasma-modified PET; *A_control_* is the optical density of supernatant obtained from fibroblasts incubated only in the culture medium.

Survival curve plotting, IC50 values calculation, and statistical analysis were performed in Excel 16.0 (Microsoft Corporation, Redmond, WA, USA) and OriginPro (version 2020b, OriginLab Corp., Northampton, MA, USA). We applied an unpaired two-tailed *t*-test for the group comparison.

#### 2.3.1. Cell Adhesion Assay

The BJ-5ta cells were serum-deprived for 2 h, and, after detachment with trypsin-EDTA solution and double washing with DMEM, were seeded into 24-well plates (60,000 cells per well in 1 mL of serum-free DMEM containing 0.1% BSA) onto the treated side of PET film (10 mm in diameter) samples and incubated under standard conditions.

Cells were allowed to adhere to the surface for 40 min, followed by four intensive washes with DMEM using a plate orbital shaker to remove non-adhered cells. After the final wash, the cells were supplemented with 1 mL of 10% FBS containing DMEM and left for 4h for complete adhesion to the membranes. Next, the standard MTT assay was performed as described in Section 2.3.

#### 2.3.2. Protein Adsorption

The protein adsorption was determined by incubating the membranes (10 mm) in 1 mL of 0.4 mg/mL protein solution in PBS pH 7.4 (BSA, fibrinogen, or lysozyme for 24 h at 37 °C under mild shaking. The protein concentration left in the supernatant was determined by applying a standard BCA protein assay after the membrane removal.

#### 2.3.3. Hemolysis Assay

The hemocompatibility of the membranes was evaluated using fresh human blood taken from healthy donors. All subjects gave their informed consent for inclusion before they participated in the study.

The blood sample was collected in tubes treated with EDTA. The red blood cell (RBC) fraction was isolated by centrifugation at 700× *g* for 10 min and washed three times with sterile PBS. Finally, the 5% *v*/*v* erythrocyte suspension was prepared in PBS.

0.785 cm^2^ polymer membranes were placed in a 24-well plate and supplemented with 0.25 mL of PBS. Then, 0.25 mL of RBC suspension was added to each well. Triton X-100 (1% *v*/*v* solution in PBS) and PBS were used as positive and negative controls, respectively. The plate was incubated for 1 h at 37 °C and centrifuged at 700× *g* for 10 min. 0.1 mL of supernatant from each well was transferred to the 96-well plate to measure the absorbance at 540 nm using a plate reader. The fraction of lyzed erythrocytes was calculated according to the following (Equation (2)):Hemolysis % = (*A_sample_* − *A_neg.control_*)/(*A_pos.controle_* − *A_neg.control_*) × 100%(2)

### 2.4. Plasma Chemical Treatment of PET

Polymeric samples were processed in the special beam-plasma chemical reactor of hybrid type, which enables to generation of sufficiently stable large-size volumes of RF discharge, electron-beam, and hybrid plasmas. Some economical estimations and power efficiency of electron-beam reactor were given in the Appendix A.

Special preliminary experiments were carried out to find the processing conditions at which the sample surface temperature did not exceed the acceptable range (about 40 °C) and polymer thermal damage did not occur. The revealed parameters are summarized in Table 1.

Figure 1a illustrates the polymer modification experiments arrangement. The HP was excited by joint action of the continuous EB and capacity coupled RF-gas discharge (13.56 MHz) on pure (without additives of any other gases) oxygen of research-grade (producer—Scientific Industrial Center of NRC “Kurchatov Institute”, Moscow, Russia). Oxygen was blown into the reaction chamber with the flow rate *G* varying within the range of 200–1000 sccm through the inlet pipeline and then staked out by the vacuum pump through the outlet pipeline (Figure 1a). As a result, the dynamic vacuum in the pressure range *P_m_* = 200–650 Pa could be obtained. The pressure control system based on the PID regulator (circled in the figure with a dotted line) automatically held the preset value *P_m_* balancing the gas intake and outtake using the butterfly valve on the vacuum pipeline and the flow controller on the inlet pipeline, respectively. 

Mass-spectrometer HALO 201-RC (Hiden Analytical, Warrington, UK) connected to the outlet pipeline continuously monitored the gas composition filling the reaction chamber. Thus, fresh oxygen continuously injected into the reaction chamber forms a pure plasma-generating medium and prevents it from possible pollution by gaseous sample destruction products. 

Oxygen has been chosen for experiments because it is widely used as an effective gas source to form oxygen-containing functional groups on various polymer surfaces [37,38,39]. In our previous studies [23] the oxygen HP was found to produce extra oxygen-containing groups in other polymers when the experiments were carried out in conditions under consideration now. Authors of these papers attributed the surface modification effect to the action of chemically active oxygen particles (primarily atomic oxygen and ozone) effectively generated in cold non-equilibrium plasmas. We will consider the HP composition and some other properties below.

The scanning EB (*E_b_* = 30 kV and *I_b_* = 1–5 mA) was injected through the mesh-type active electrode into the gap of the planar double-electrode RF discharge system. Electromagnetic deflecting coils operated by the electron-beam sweep controller EBS-530 (Inficon Inc., East Syracuse, NY, USA) scanned the EB in the {*x*, *y*} plane. The PET films to be treated were placed in the gap between the active RF-electrode and the flat passive electrode that usually was grounded. In experiments on the EBP modification, the RF discharge was switched off and vice versa the EB was not injected into the reaction volume when the polymers were treated in simple RF discharge.

The gas flow rate and the pressure of plasma-generating gas were automatically controlled by mass flow and pressure controllers (MKS Instruments, Milpitas, CA, USA). The sample heating was remotely monitored with an Optris LS IR-pyrometer (Optris, Berlin, Germany). The pyrometer was equipped with a light filter transparent for IR but absorbing visible optical radiation to separate long wavelength thermal radiation of the polymeric materials (wells and samples) from their luminescence in plasmas. The required process temperature was maintained constant and adjusted by the electron beam current *I_b_*.

PET samples were placed at the wells bottom of 24-well plates (Corning TCT surface treatment, Lowell, MA, USA) for plasma chemical treatment (Figure 1b). Immediately after the plasma modification, the plates with the treated PET samples were packed in sterile plastic bags (NPO Vinar, Moscow, Russia).

### 2.5. Physical Aspects of PET Modification in Beam-Plasma Chemical Reactor

From the point of view of physics and technology, the experiment arrangement was as follows. The plasma cloud should be excited inside a well in a dielectric container to generate fluxes of chemically active particles able to controllably modify the cavity’s inner surface without thermal damage it. The modification effect was supposed to depend on the flux intensity and particle activity. The cold plasma of oxygen is a source of such particles, and the HP generation technique is effective in forming intensive particle fluxes. When an EB and gas discharge jointly ionize and excite plasma generating medium their combination realizes advantages of each of them: the beam is an effective ionizer whereas the gas discharge produces excited particles better than the beam.

The beam was proven to control the plasma cloud location [24]: the cloud is concentrated around the beam and follows it when the beam scans in space. Thus, the plasma cloud controlled by the beam can be considered like a “plasma brush” and creates complicated patterns on samples with distributed cavities. Besides, the beam transports heavy plasma particles into the cavity: being generated outside the cavity by the gas discharge they can be delivered into the cavity by the EB. Thus, the RF discharge is unnecessary inside the cavity, which avoids several well-known problems of discharge plasma interaction with solid surfaces including sheath. The problem of the electrostatic charging of dielectrics by electron beam irradiation can be settled by gas pressure increase. Our experiments with the EBP near the surface have shown that the potential of any body surrounded with or contacting the EBP does not exceed floating plasma potential (about several V) if gas pressure reaches 200–650 Pa (depending on the plasma generating gas composition) [24].

Figure 1b illustrates the HP-processing procedure of the Corning well plate with PET samples. The sample illuminates homogeneously and no dark zones or electrical breakdowns are detected. Nevertheless, some information about plasmas characteristics and properties is worth to be given.

To describe and analyze the EBP and HP generation and their interaction with the solid body the following physical model was used. Originally a thin monochromatic electron beam is injected into still oxygen bulk. The beam trajectories have some initial angular spread. Electrons take part in multiple collisions both elastic and inelastic. As a result, the beam is scattered and decelerated, losing energy from the original value to the thermal one. This forms the degradation spectrum of electrons. Gas molecules are ionized and excited by electron impact in inelastic collisions producing numerous secondary electrons and heavy particles both charged and neutral.

Electron energy distribution functions in the EBP and, consequently, in the HP are non-Maxwellian and have a long tail of electrons of intermediate energy. Partially decelerated original electrons together with energetic secondary ones are able to take part in other inelastic processes again and again. Note that the electron energy distribution function of gas discharge is also non-Maxwellian and also has a tail but is significantly shorter than that in the EBP. Finally, the original beam power transforms into thermal energy or radiates.

When the EB is injected into a closed gas volume filling a container the scattered electrons heat the container wall by bombardment and then the wall heats the gas by heat transfer. As a result, some spatial profiles of temperature, gas density, plasma particle concentrations and local release of the beam power form are formed. Heavy plasma particles move toward the container wall due to the concentration and thermal diffusions (neutral particles) or ambipolar diffusion (electrons and positive ions). Reaching the wall they can modify container wall material chemically. Being bombarded directly by fast electrons the wall can accumulate the electrostatic charge. From this point of view, it is particle fluxes that are the most important quantities.

Figure 2 presents fluxes (*q*) of electrons, atomic oxygen O, oxygen ions O_2_^+^ and excited oxygen O_2_(a) falling on the cylindrical container bottom at a different distance from its centre. The EB was injected along the container axis and the injection point was at a distance of 150 mm from the bottom.

The fluxes were calculated in numerical simulations. There are two approaches to computer HP simulation: (a) the EB is supposed to be injected into the preliminary excited plasma of RF discharge, and (b) an external electromagnetic field is superimposed on the EBP volume. The first approach demands information about gas discharge. Many authors carried out simulations for conditions close to those of our experiments [40,41]. Usually, numerous plasma processes are taken into account. The second approach seems to be simpler because the sufficiently restricted number of reactions (about 40) are necessary for oxygen HP description. Moreover, for qualitative description with acceptable accuracy, only six reactions may be used (Table 2). Also we calculated longitudinal profiles of gas and container wall temperature, fluxes of plasma particles and release of the beam power to predict possible modification effect for various zones of the container wall and estimate risks of sample thermal damage. Simulations were carried out by Monte-Carlo method on the basis of continues deceleration model for all electrons of degradation spectrum excluding thermolized ones. The example of this simulation is presented in Figure 3 for model dielectric cavity 18 mm in diameter. The beam characteristics and gas pressure indicated in caption for Figure 3 correspond with real conditions of experiments under consideration data of which are useful to verify physical model and simulation algorithms used.

Simulations and preliminary physical experiments gave information to optimize treatment procedures for real samples. In particular, the best distances *z* from the beam injection window to the well front plane at selected oxygen pressures *P_m_* and beam currents *I_b_* were found to protect samples from overheating and to avoid electric charge accumulation. Under optimized conditions, no thermal damages and effects caused by electrostatic phenomena were observed. In real experiments, the sample temperature was continuously monitored by an optical pyrometer. Since it detects not only sample thermal radiation but also sample visible luminescence (Figure 1b) special light filter transparent for IR radiation was recommended for usage. Contact temperature measurements were difficult because thermocouples and thermo resistor signals were dramatically distorted by noises from the gas discharge and, to some lesser extent, from the EB. Being carried out after the treatment stopped the contact temperature measurements were useful for optical pyrometer calibration. As to electrostatic phenomena, it was found that at plasma pressures higher than 200 Pa polymers were fully discharged via the EBP.

## 3. Results

### 3.1. Morphology and Chemical Composition of the NTPs-Modified Polymers

Figure 4 shows typical large-scale AFM images (50 µm × 50 µm) of the original and plasma-modified PET surfaces where considerable topographical changes caused by the plasma exposure can be observed. Being etched by the highly reactive oxygen plasma species the HP-treated polymeric surface acquired an uneven granular structure (Figure 4d) whereas the EBP-modified PET remained smooth and uniform (Figure 4c). The latter effect can be explained by the fact that high-energy electrons of the EB provoke the crosslinking of polymeric chains which results in some smoothing of the surface texture.

AFM images were analyzed in terms of average surface roughness (*R_a_*). The roughness values are 16.10 ± 3.1 nm for original PET and 20 ± 2.0 nm, 10,9 ± 2.5 nm and 38.8 ± 2.7 nm for PET modified in RF discharge, EBP and HP, respectively.

The elemental composition and relative concentrations of C and O atoms on the PET surface before and after plasma chemical treatment are presented in Table 3. The C/O ratio in the PET surface layer decreased from 3.56 (control sample) to 1.61, 1.47, and 1.46 after modification by the RF discharge, electron, and hybrid plasmas, respectively. Thus, the significant rise of the oxygen content with the simultaneous reduction of carbon concentration was revealed in all plasma-modified PET films.

In the untreated PET sample, the C1sA peak (attributed to carbon atoms of the aromatic ring) accounted for about 59% of the total amount of carbon, while the C1sB and C1sD peaks (which characterize carbon atoms in the methylene and ester groups) accounted for 8.43% and 8.37%, respectively. After plasma processing, the area of the C1sA peak decreased, whereas the C1sB and C1sD peaks areas increased in all PET samples. These data suggest the formation of additional oxygen-containing functional groups on polymer surfaces due to plasma chemical treatment. The intensity of oxidation depended on the type of NTP applied for the PETs modification: the most significant effect was observed after PET treatment in the RF discharge and hybrid plasmas.

Figure 5 presents the C1s XPS-spectra of PET films before and after their treatment in NTPs. Again, three characteristic peaks with binding energies (BE) of 283.57 eV (C–C bonds of the aromatic ring), 285.13 eV (C–O) and 287.54 eV (O=C–O) were identified in the original PET sample. After the HP-plasma modification the area of the C1sD (O–C=O) peak, which is attributed to the carboxyl groups, increased by 35% while the area of the C1sA peak associated with methyl and methylene carbons decreased by 25% in comparison with the original sample. At the same time, the rise of the peaks corresponding to the C–O (O1sA) and C=O (O1sB) groups was found in the O1s XPS spectra of the plasma-treated PET with the most significant changes after RF discharge and HP action (Figure 6). Besides, an additional peak (peak C1sC in Figure 5b–d), that may belong to the carbon of the C=O functional group [32,33,34], is observed in XPS spectra of plasma-modified PET.

Thus, the XPS results indicate that the carbon bonds on the polymer surface are destroyed during the plasma treatment and then they react with the oxygen species (atomic oxygen, OH radical, etc.) produced in plasma, forming carbonyl and carboxyl groups.

The results of the FTIR spectroscopy confirm the formation of oxygen-containing functional groups. The most significant changes in the IR spectra of PET films were found in the wavenumber range of ν = 3500–3100 cm^−1^, in which a broad absorption band appeared, characterizing the stretching vibrations of O–H bonds. An increase in the intensity of the peak at ν = 1686 cm^−1^ (C=O stretching vibrations in carboxyl groups) was also noted. A higher concentration of these groups was detected in HP-treated films (Table 4).

However, reduced intensity of the absorption band at ν = 1712 cm^−1^ attributed to C=O stretching vibrations of ester groups which are parts of the elementary units of PET chain was observed in plasma-treated samples. The explanation for such a decrease may be the destruction of PET chains together with the formation of vinyl derivatives of the esters (CO–O–C=C) [42]. The data presented in Table 4 also suggest the formation of double bounds due to the plasma chemical treatment that was more intensive after PET modification in EBP and HP.

### 3.2. Hydrophilic Properties of NTPs-Modified PET Films

Contact angles *θ_W_* and *θ_DM_* measured with water and diiodomethane, respectively, the calculated values of the total surface energy *γ_tot_*, as well as its polar *γ_pol_* and dispersive *γ_disp_* components are given in Table 5. The increases in *γ_tot_* from 42.7 mJ/m^2^ to 70.4 mJ/m^2^ and 71.6 mJ/m^2^ were achieved after PET modification in the RF discharge plasma and HP, respectively. The polar component became more effective while the contribution of *γ_disp_* decreased, suggesting the formation of oxygen-containing polar chemical groups at the plasma-modified PET surfaces that was consistent with the FTIR and the XPS data.

### 3.3. The Effects of NTPs-Modified PET Films on Cell Attachment and Growth

The hybrid plasma treatment resulted in oxygen-containing polar hydroxyl, carbonyl, and carboxyl groups’ formation on the PET sample’s surface. We also observed surface free energy increase and wettability enchantment up to 1.5–2 times in comparison with the original PET. The changes in the chemical composition of the polymeric surface together with the increment of its hydrophilic properties result in the improvement of compatibility of the plasma-modified PET films with living cells and tissues.

The biological tests of the plasma-treated PET samples were performed on the BJ-5ta line of immortalized human fibroblasts cell culture, which is commonly used for assessment of the cytotoxicity and biocompatibility of the new materials.

The microscopic images of the fibroblasts spreading on the PET samples modified in oxygen hybrid plasma can be found in Figure 7. The cells exhibited a flattened morphology and good adherence to the polymeric surface. The normal cell morphology and similar proliferation patterns among all samples evidenced a lack of pronounced cytotoxicity and negative effects of the treated PET surfaces on cell culture.

We confirmed the lack of pronounced cytotoxic effect of PET modified by oxygen hybrid plasma by MTT assay. The observed survival rates in wells containing the samples were lower than in control wells during the whole incubation, which could be explained by special treatment promoting cell adhesion, while a significant part of the wells bottom contained PET films was covered. This survival difference increased with the incubation period prolongation.

After 48 h, according unpaired two-tailed t-test, we observed a significant difference between untreated PET and other samples (untreated PET vs. RF discharge 25 W *p* = 0.0034; untreated PET vs treated in RF discharge 35 W *p* = 0.0073; untreated PET vs. EBP-modification *p* = 0.0002), except treatment in hybrid plasma, according to unpaired *t*-test. After 72 h, only the EBP demonstrated significantly lower cell survival compared with untreated PET (*p* = 0.0019). After 96, surprisingly, only the EBP sample revealed significantly higher survival compared with untreated PET (*p* = 0.0482) (Figure 8).

After 96 h of incubation, all samples demonstrated comparable survival results—total survival was in the 65–75% range of control. Overall, the data obtained demonstrated a lack of pronounced cytotoxicity and high biocompatibility of plasma-treated PET samples.

#### 3.3.1. Cell Adhesion Assay

Interactions between cells and the extracellular matrix (ECM), including polymer membranes, can be a prerequisite for the design of new biocompatible artificial materials, the development of new monitoring techniques and cell survival in new microenvironments [43]. Thus, considering relatively interesting results of survival analysis after direct cells seeding onto membranes, further, we examined the cell adhesion at PET membranes subjected to various types of treatment during the first 40 min after seeding (Figure 9).

We did not notice any notable differences during the microscopic examination of the cell culture (Figure 9a). However, in contrast with the initial results of the MTT test obtained after direct seeding the cells onto membranes—the well washing revealed significant diversity in cell adhesion to plasma-treated membranes during the initial phase of incubation. The differences were considered significant (unpaired two-tailed *t*-test with Welch’s correction): untreated PET vs treatment in RF discharge (*p* < 0.0001); untreated PET vs EBP-treatment (*p* = 0.01); untreated PET vs treatment in hybrid plasma (*p* = 0.0013).

#### 3.3.2. Protein Adsorption

Noticing the influence on the initial stages of cell adhesion, we evaluated protein adsorption on the membranes by applying indirect BCA analysis (Figure 10).

The intact PET membranes displayed reduced protein adsorption compared to plasma-treated samples. A significant difference was observed for RF discharge, EBP, and hybrid plasma-treated samples incubated in solutions of all tested proteins. We observed a lack of significant difference among treated samples (Figure 10).

The observed tendency of higher lysozyme absorption onto all samples is explained probably by electrostatic interactions of positively charged protein.

#### 3.3.3. Hemolytic Activity

Next, we performed hemolytic activity analysis to evaluate the non-specific toxicity of the membranes. Generally, the hemolysis rate is described by HC_50_ value—the concentration of the drug or formulation causing 50% RBCs lysis upon 1 h exposure [44]. The samples characterized with a lysis rate of less than 5% are generally considered non-haematotoxic. In our case, it was impossible to evaluate mass-dependent hemolytic activity, so we adapted the conventional protocol and incubated plasma-treated or control membranes in the presence of RBCs.

The results presented in Table 6 evidence significantly higher untreated PET hemolytic activity in contrast with plasma-treated membranes with *p*-values less than 0.05. Probably, this effect can be explained by the pronounced hydrophobic properties of untreated PET surface, which are negated after treatment with plasma [45]. The RBCs membrane destruction is generally caused by the hydrophobic interactions between the hydrophobic groups of polymers and the lipid bilayer [46].

## 4. Discussion

It is well known that NTPs affect polymeric films in several ways, the most important of them are: (1) cleaning from organic impurities (2) etching by removal of material from the surface and (3) producing functional groups because of surface reactions stimulated by chemically active plasma particles. The etching process alters the surface topography enhancing its roughness, which has been detected after PET modification HP-modification by AFM. These changes lead to the polymer wettability improvement without modifying its surface texture. On the other hand, atomic oxygen, other neutral oxygen-containing radicals, and numerous exited particles formed in the HP react with polymers to produce a variety of polar functional groups (such as –OH, –COH, C=O, and –COOH), which causes the rise in the surface hydrophilicity as well. The results of XPS analysis suggest the significant oxidation of PET polymeric chains during the NTPs treatment, which was the most intensive in the case of RF discharge and hybrid plasmas.

The major reactive species responsible for the incorporation of oxygen-containing polar groups into the polymer surface are believed to be atomic oxygen [9,10,47]. These neutral active oxygen particles react with the dangling bonds on the PET surface which are formed due to C–C bond breaking and lead to the oxidation of the treated surface. The dominant initial step can be the hydrogen abstraction by the O atom of organics RH, i.e., RH  ±  O  →  •R  ±  OH [48]. The possible following reactions are given in Figure 11 [49].

Both etching and plasma-stimulated oxidation result synergistically in the polymer hydrophilicity increase, which promotes stable cell adhesion and healthy cellular functioning. The higher roughness formed after plasma processing also provides the focal adhesion points for cell interactions with the polymeric surface. At the same time, the oxygen-containing groups control the essential protein adsorption from the cell surface onto the PET thus enhancing the bonding of cell integrins with the polymeric substrate [50]. In vitro analysis revealed good biocompatibility of the treated samples. Visually the cell morphology on the surfaces was similar among all samples and control cells seeded onto tissue-treated polystyrene plate bottom. Th survival assay revealed some differences increased during incubation—after 48 h incubation cells cultured onto plasma-treated surfaces demonstrated 75–85% survival (95% for untreated PET); after 72 h survival rates were in the 75–90% range; and after 96 h we observed 65–75% survival in contrast with control cells. The observed differences in survival among samples could be explained by a range of factors, for instance, slight variation in the chemical structure of poly(ethylene terephthalate samples which were used for NTPs processing or micro-heterogeneity (non-genetic cell to cell variability) of fibroblasts [51,52]. Therefore, plasma-treated PET samples demonstrated good overall biocompatibility notwithstanding the tendency to slower cell expansion over the polymeric surface.

Despite implicit results after direct cells seeding onto membranes—further analysis revealed some differences: short initial incubation followed by washing revealed a statistically significant difference between cells’ attachment degree onto untreated PET and plasma-treated samples—the cells attached better onto treated membranes. Adsorption analysis displayed better protein binding efficacy of plasma-treated membranes in contrast with untreated PET. Finally, the plasma-treated membranes revealed lower hemolytic activity. The last in vitro results were most probably explained by surface charge contributed by the plasma treatment—the charged surface displayed better protein adsorption and higher cell adhesion properties evidencing well potential biocompatibility of these samples and prospects of cold-plasma application in polymer surface treatment.

At the same time, using the EB it is possible to precisely localize the RF discharge on the desirable zone of the polymeric sample surface, maintain the set temperature in this zone, and control plasma chemical functionalization processes. As a result, areas within which physical, chemical, and functional properties change either abruptly (structured patterns) or smoothly (gradient materials) can be formed on the surface. Experimental evidence confirming these options’ feasibility can be found in our papers [23,24].

## 5. Conclusions

The applicability of beam-plasma chemical reactors generating cold hybrid plasma for the production of noncytotoxic polymeric surfaces with high hydrophilicity and good biocompatibility with human fibroblast culture and human red blood cells was studied. Oxygen hybrid plasma was exited by joint action of a continuous scanning electron beam and a capacity-coupled RF-gas discharge (13.56 MHz). Hybrid plasma treatment caused polar oxygen-containing functional groups formation in the surface layer of poly(ethylene terephthalate) films. No thermal or radiative damage in tested polymer samples were found. The plasma-modified polymers turned out to be noncytotoxic and revealed good biocompatibility with human fibroblasts BJ-5ta as well as lower hemolytic activity than untreated poly(ethylene terephthalate).

Experiments demonstrated that in hybrid plasma the problem of the electrostatic charging inherent in polymer treatments in gas discharges can be settled because the electron beam component of hybrid plasma eliminates the charge accumulated on polymeric film. Moreover, the electron beam characteristics, namely scanning mode and power, can instantly and independently control the reaction volume geometry as well as concentrations of active plasma particles. As a result, the RF discharge can be accurately localized at the desirable zone of the polymeric surface or even inside cavities. This provides new approaches to the production of abruptly structured patterns or smooth gradients of functionalities on a plane and 3D polymeric items of complicated geometry without special masks or covers conventionally used in gas-discharge plasma technologies.

## Figures and Tables

**Figure 1 polymers-16-00172-f001:**
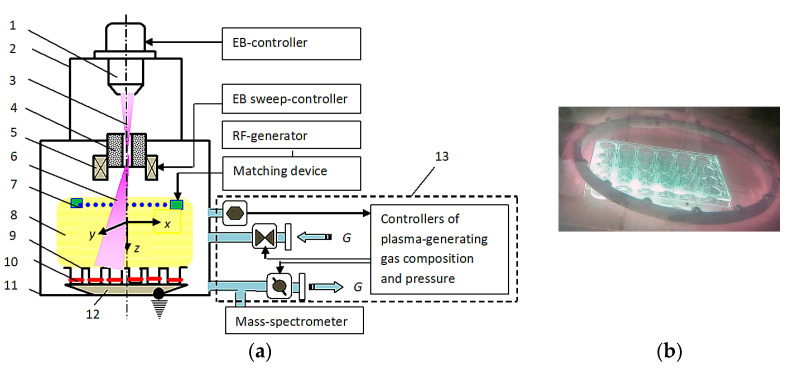
Principle scheme of the experimental setup (**a**) and samples treatment procedure (**b**): 1—electron gun, 2—high-vacuum chamber, 3—electron beam, 4—injection window, 5—injection window, 6—electron-beam plasma, 7—active mesh-type RF-electrode, 8—RF-plasma, 9—polymer container, 10—working chamber, 11—polymer samples, 12—container holder (grounded passive RF-electrode), 13—PID-regulator.

**Figure 2 polymers-16-00172-f002:**
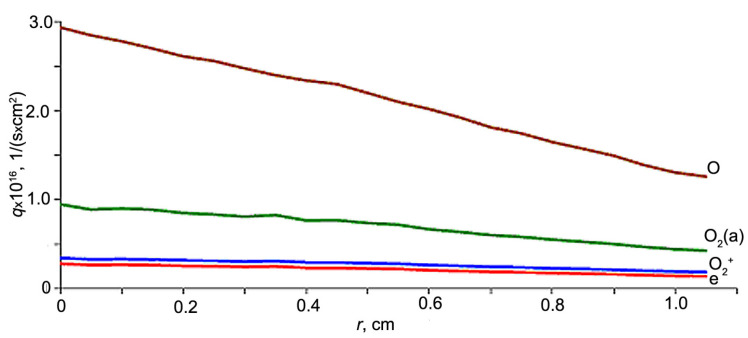
Fluxes (*q*) of electrons, atomic oxygen O, oxygen ions O_2_^+^ and excited oxygen O_2_(a) falling on the cylindrical container bottom: *E_b_* = 30 keV, *I_b_* = 10 mA, *P_m_* = 650 Pa; *r*—radial distance from the bottom center.

**Figure 3 polymers-16-00172-f003:**
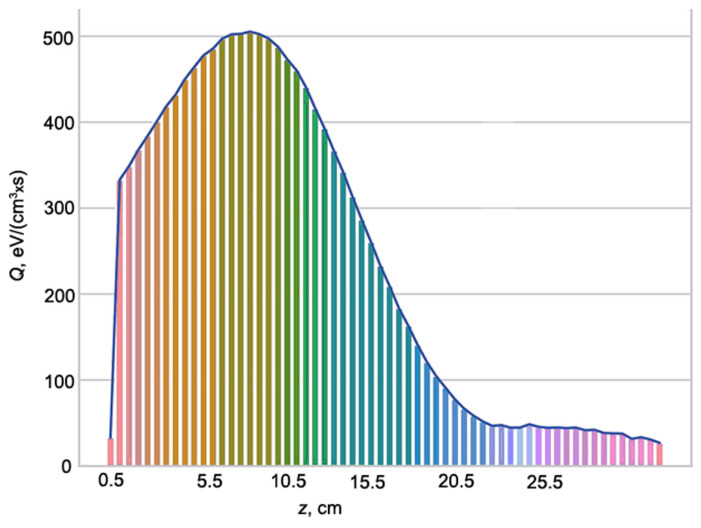
Longitudinal profiles of the beam power release (*Q*) in the model cavity (polystyrene tube) 18 mm in diameter: *E_b_* = 30 keV, *I_b_* = 2 mA, *P_m_* = 650 Pa.

**Figure 4 polymers-16-00172-f004:**
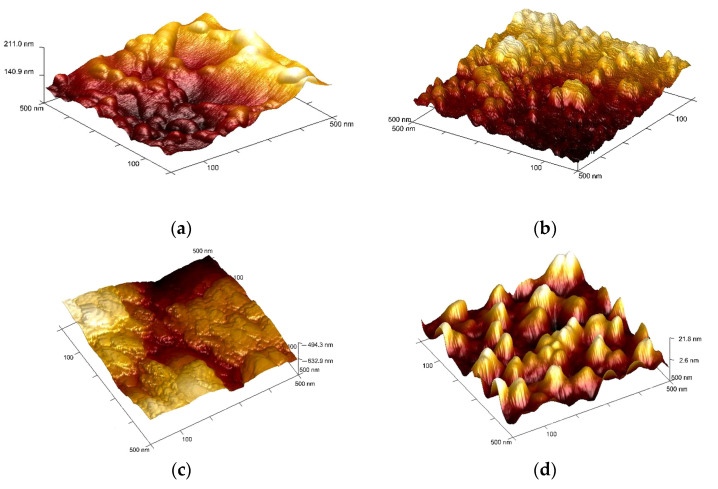
AFM images of (**a**) original PET surface and PET surfaces treated in (**b**) RF-discharge plasma; (**c**) Electron-beam plasma; and (**d**) Hybrid plasma. Treatment conditions: *U* = 30 kV, *I_b_ =* 2 mA, *N_RF_* = 35 W, EB scanning speed = 150 ms, EB spin speed = 25 rpm, *P_m_ =* 650 Pa, τ = 5 min.

**Figure 5 polymers-16-00172-f005:**
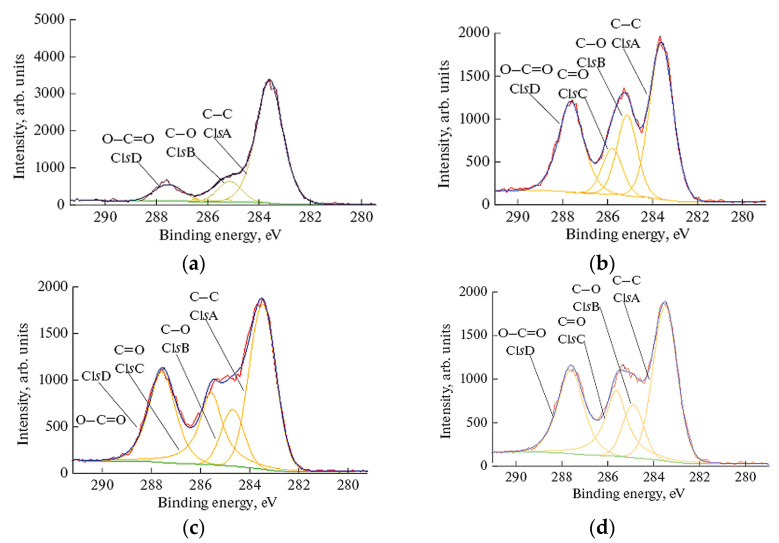
C1s XPS spectra of PET samples before and after plasma treatment: (**a**) untreated PET; (**b**) PET treated in RF discharge plasma; (**c**) PET treated in electron-beam plasma; (**d**) PET treated in hybrid plasma. Treatment conditions: *U* = 30 kV, *I_b_* = 2 mA, *N_RF_* = 35 W, EB scanning speed = 150 ms, EB spin speed = 25 rpm, *P_m_* = 650 Pa, τ = 5 min.

**Figure 6 polymers-16-00172-f006:**
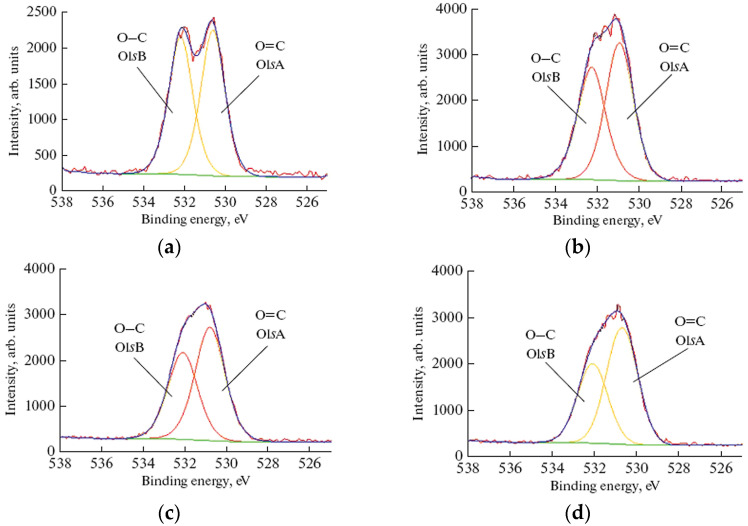
O1s XPS spectra of PET samples before and after plasma treatment: (**a**) untreated PET; (**b**) PET treated in RF discharge plasma; (**c**) PET treated in electron-beam plasma; (**d**) PET treated in hybrid plasma. Treatment conditions: *U* = 30 kV, *I_b_* = 2 mA, *N_RF_* = 35 W, EB scanning speed = 150 ms, EB spin speed = 25 rpm, *P_m_* = 650 Pa, τ = 5 min.

**Figure 7 polymers-16-00172-f007:**
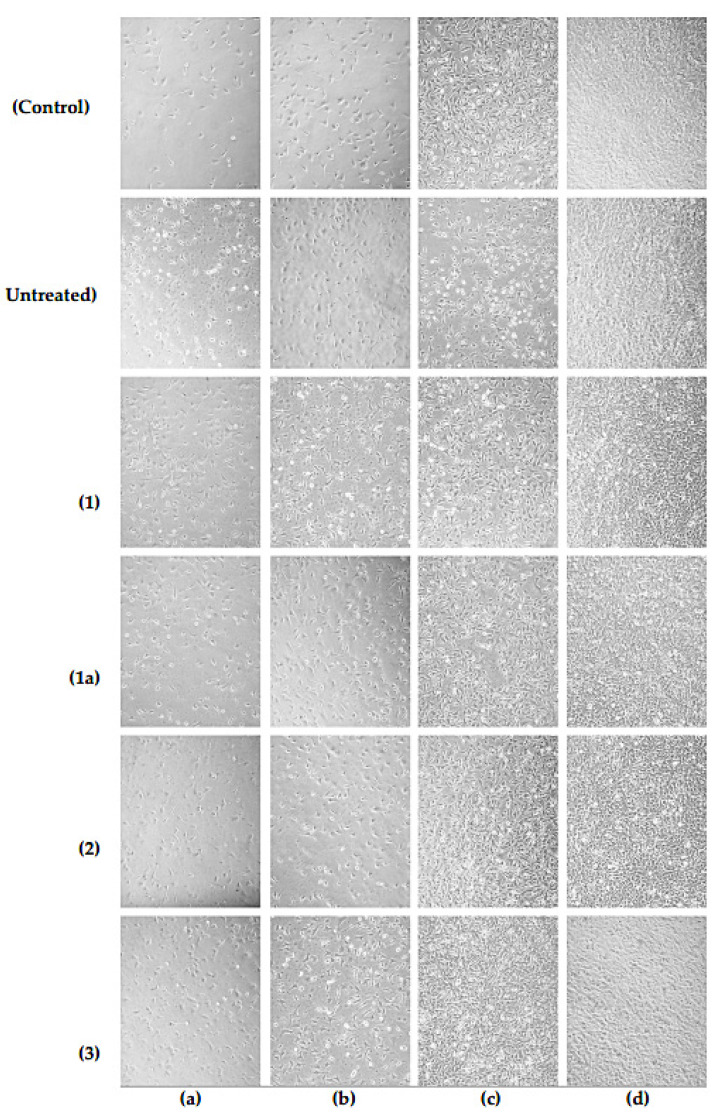
The optical microscopic images of the BJ-5ta fibroblasts after 24 (**a**), 48 (**b**), 72 (**c**) and 96 h (**d**) of their incubation on: (control)—polystyrene plate bottom; (untreated)—untreated PET; (1)—RF discharge (25 W); (1a)—RF discharge; (2)—EBP; (3)—hybrid plasma. Treatment conditions: *U* = 30 kV, *I_b_* = 2 mA, *N_RF_* = 35 W, EB scanning speed = 150 ms, EB spin speed = 25 rpm, *P_m_* = 650 Pa, τ = 5 min.

**Figure 8 polymers-16-00172-f008:**
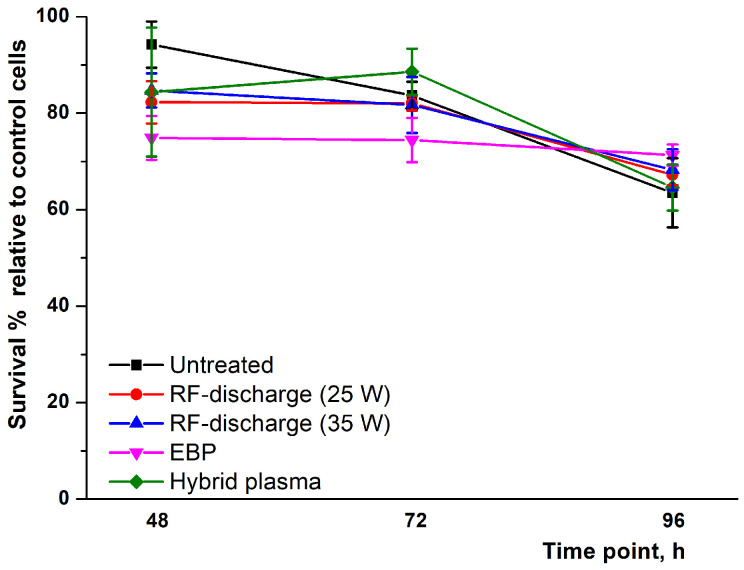
Mean values of the BJ-5ta fibroblast survival cultivated on PET films during 48, 72 and 96 h in 24-well culture plates in comparison with the average survival value of control cells cultured in polystyrene tissue traded wells in the absence of PET samples. n = 5. Treatment conditions: *U* = 30 kV, *I_b_* = 2 mA, *N_RF_* = 25 W or 35 W, EB scanning speed = 150 ms, EB spin speed = 25 rpm, *P_m_* = 650 Pa, τ = 5 min.

**Figure 9 polymers-16-00172-f009:**
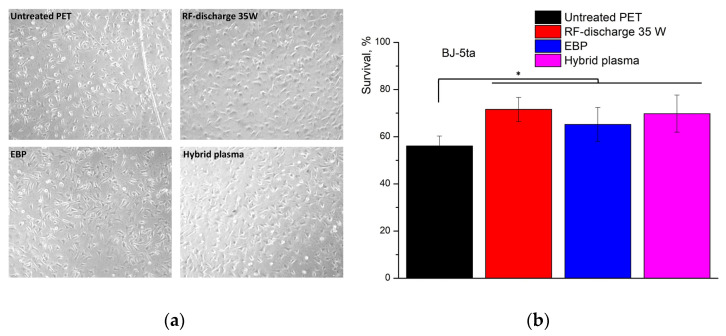
BJ-5ta cells attachment to differentially treated PET membranes. (**a**)—representative images of cell culture just before MTT analysis; (**b**)—relative survival values of cell culture seeded onto different PET membranes. * *p* < 0.01, n = 8. Treatment conditions: *U* = 30 kV, *I_b_* = 2 mA, *N_RF_* = 35 W, EB scanning speed = 150 ms, EB spin speed = 25 rpm, *P_m_* = 650 Pa, τ = 5 min.

**Figure 10 polymers-16-00172-f010:**
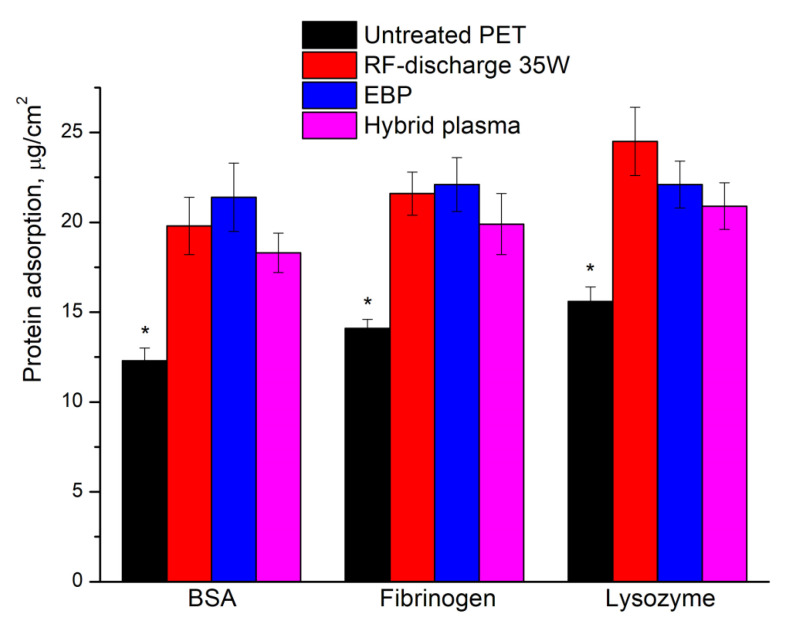
Protein adsorption was evaluated by BCA assay. * *p* < 0.01, n = 8. Treatment conditions: *U* = 30 kV, *I_b_* = 2 mA, *N_RF_* = 35 W, EB scanning speed = 150 ms, EB spin speed = 25 rpm, *P_m_* = 650 Pa, τ = 5 min.

**Figure 11 polymers-16-00172-f011:**
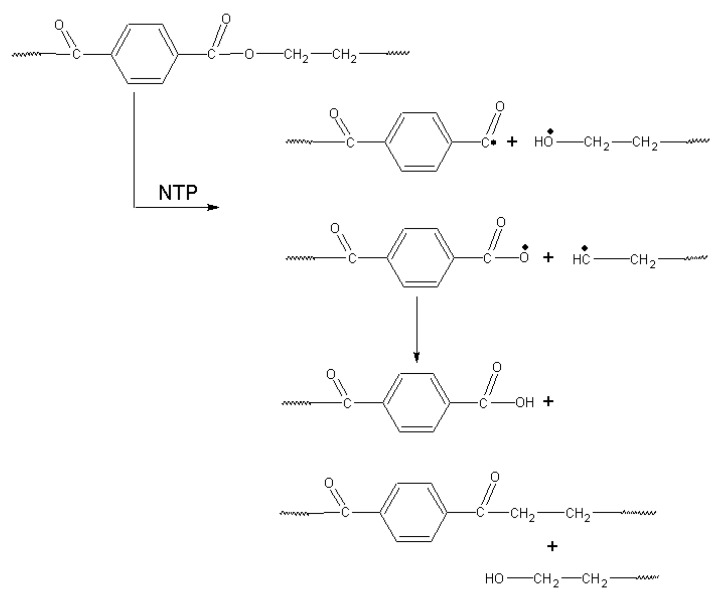
The possible reactions occurring during PET plasma chemical treatment [49].

**Table 1 polymers-16-00172-t001:** The treatment conditions are used for polymer modification in various types of NTPs.

Parameter	Value
Accelerating voltage (*U*)	30 kV
EB current (*I_b_*)	1–5 mA
Electron beam scanning mode	Concentric circles with a diameter of 10 cm
Electron beam scanning speed	140–150 ms
Spin speed of electron beam shape rotation	20–25 rpm
Plasma-generating gas pressure (*P_m_*)	200–650 Pa
Oxygen flow rate (G)	200–1000 sccm
RF discharge power (*N_RF_*)	25–35 W
Treatment time (τ)	5 min
Material temperature (*T_s_*)	40 °C

**Table 2 polymers-16-00172-t002:** The kinetic scheme is used for the calculation of neutral particle concentration in the oxygen HP.

Reaction	The Energy (eV (*)) or the Rate Constant (cm^3^ × s^−1^ or cm^6^ × s^−1^ (**))
e_b_ + O_2_ → e_b_ + O + O	20.2 (*)
e_b_ + O_2_ → e_b_ + O_2_(a)	23.1 (*)
O + 2O_2_ → O_3_ + O_2_	6.9 × 10^−34^ (300/*T_g_*)^1.25^ (**)
O + O_3_ → O_2_ + O_2_	2 × 10^−11^ exp (−2300/*T_g_*)
O + O + O_2_ → 2O_2_	6.7 × 10^−33^ (300/*T_g_*)^0.63^ (**)
O_3_ + O_2_ → O + 2O_2_	1.65 × 10^−9^ exp (−11,400/*T_g_*)

**Table 3 polymers-16-00172-t003:** C1s and O1s binding energies and surface elemental composition of PET before and after plasma chemical treatment.

Peak		Original PET		PET Treated in RF Discharge		PET Treated in EBP		PET Treated in Hybrid Plasma	
		BE, eV	Content of the Element, %	BE, eV	Content of the Element, %	BE, eV	Content of the Element, %	BE, eV	Content of the Element, %
C1s	A	283.57	59.19	283.55	23.19	283.49	25.72	283.51	24.81
	B	285.13	8.43	285.17	13.08	284.69	7.71	284.89	7.44
	C	–	−	285.94	6.80	285.57	13.18	285.63	12.40
	D	287.54	8.37	287.60	16.30	287.56	15.20	287.60	15.09
O1s	A	530.71	13.92	530.88	18.96	530.87	19.23	530.84	19.89
	B	532.32	7.44	532.18	21.67	532.18	18.97	532.18	20.38

BE—binding energy. Treatment conditions: *U* = 30 kV, *I_b_ =* 2 mA, *N_RF_
*= 35 W, EB scanning speed = 150 ms, EB spin speed = 25 rpm, *P_m_ =* 650 Pa, τ = 5 min.

**Table 4 polymers-16-00172-t004:** Ratios of the IR-absorbances (*D*) at different wavenumbers for PET before and after plasma chemical treatment.

Sample	*D*_3250_/*D*_1500_ O–H	*D*_1776_/*D*_1500_ CO–O–C=C	*D*_1712_/*D*_1500_ C=O (Ehter Group)	*D*_1686_/*D*_1500_ C=O (Carboxyl Group)	*D*_1608_/*D*_1500_ C=C
Original PET	0.140	0.040	7.172	2.068	0.528
PET treated in RF discharge	0.106	0.092	6.778	2.644	0.633
PET treated in EBP	0.243	0.254	7.215	3.519	1.068
PET treated in hybrid plasma	0.267	0.192	7.248	2.852	0.877

The peak with *ν* = 1500 cm^−1^ corresponds to the maximum absorption band attributed to C–H vibrations in the benzene ring. Treatment conditions: *U* = 30 kV, *I_b_* = 2 mA, *N_RF_* = 35 W, EB scanning speed = 150 ms, EB spin speed = 25 rpm, *P_m_* = 650 Pa, τ = 5 min.

**Table 5 polymers-16-00172-t005:** Contact angles, SFE, and SFE components for PET films before and after modification in oxygen NTPs.

Sample	*θ_W_*, °	*θ_DM_*, °	*γ_pol_*, mJ/m^2^	*γ_disp_*, mJ/m^2^	*γ_tot_*, mJ/m^2^
Control	80.6 ± 0.2	40.2 ± 1.5	3.3	38.4	42.7
RF discharge treated	27.3 ± 0.6	36.3 ± 1.8	30.1	40.2	70.4
EBP-treated	44.9 ± 0.1	42.8 ± 0.3	19.1	45.8	64.9
HP-treated	26.8 ± 0.4	36.8 ± 0.2	30.3	41.3	71.6

The data are presented as mean ± standard deviation. All the results are significant compared to control (untreated polymer) (*p* < 0.05). Treatment conditions: *U* = 30 kV, *I_b_* = 2 mA, *N_RF_* = 35 W, EB scanning speed = 150 ms, EB spin speed = 25 rpm, *P_m_* = 650 Pa, τ = 5 min.

**Table 6 polymers-16-00172-t006:** Hemolytic activity of different PET membranes.

Sample	Hemolytic Activity, %
Tritox X-100, 1% PBS	100 1.7 ± 0.5
Untreated PET, ~3.14 cm^2^	7.1 ± 0.6 *
RF discharge 35 W, ~3.14 cm^2^	3.2 ± 0.6
EBP, ~3.14 cm^2^	3.9 ± 0.9
Hybrid plasma, ~3.14 cm^2^	4.2 ± 0.7

* *p* < 0.05, n = 8 Treatment conditions: *U* = 30 kV, *I_b_* = 2 mA, *N_RF_* = 35 W, EB scanning speed = 150 ms, EB spin speed = 25 rpm, *P_m_* = 650 Pa, τ = 5 min.

## Data Availability

Data are contained within the article.

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
