# Peer review of "Applicability of Electron-Beam and Hybrid Plasmas for Polyethylene Terephthalate Processing to Obtain Hydrophilic and Biocompatible Surfaces"

_polymers, 2024, doi:10.3390/polym16020172_

Round 1
Reviewer 1 Report
Comments and Suggestions for Authors
In the manuscript “Applicability of Electron-beam and Hybrid Plasmas for Polyethylene Terephthalate Processing to Obtain Hydrophilic and Biocompatible Surfaces”, the authors discussed the effect of various plasma treatment configurations to improve the biocompatibility of polyethylene terephthalate (PET). Plasma from atmospheric to high vacuum systems is routinely used to modify polymeric surfaces for applications such as enhancing biocompatibility and improving cell adhesion and proliferation. The present research did not show any substantial novelty worthy to be published. However, the authors may continue to improve the manuscript by addressing the following:
- In the abstract, what is meant by “...according to desirable law.”?
- During the hemolysis assay, was the donor informed about using his/her blood for the study?
- For the plasma system and treatment process, it would be more informative if the dosage and current density of the electron beam were reported rather than just the current. What is the electron beam spot size? How about the electron energy arriving at the PET surface? Are the electrons monoenergetic? Are the electrons thermalized or not? At 100s Pa pressure, what will be the interactions of the electrons with the oxygen molecules? Regarding the material temperature, how and where was the temperature measured? Energetic electrons can easily heat up the surface locally. The authors need to verify electron beam-induced heating.
- In Table 1, the authors showed varying some of the operating parameters. However, in the results, the authors did not mention the exact parameter used when reporting their results. For instance, in Fig. 2, what was the operating pressure used to treat the PET?
- Indicate the purity of oxygen used.
- In the surface free energy calculations, it seems that there is no significant difference between RF, EBP, and HP. Significance is only with the untreated polymer which is expected. How do the authors differentiate the treatment processes?
- In Fig. 6, the y-axis should start at zero.
- In Fig. 7, there is no significance in the survival rate.
- The authors should also check the presence or enhancement of hydroxyl groups on the surface. If not with XPS, the authors can try FTIR.
- One of the major interactions overlooked by the authors is the effect of the plasma sheath formed on the surface. When non-conductive polymers are exposed to a discharge, a sheath if form which may have different properties with the bulk plasma. The authors need to verify how the sheath affects the surface condition. RF-induced plasma has different sheath dynamics as compared to EB only. However, the relatively high-pressure conditions can complicate the collisions inside the sheath.
Comments on the Quality of English Language
Use of the English language can still be improved.
Author Response
Dear Reviewer,
Thank you very much for the Referee’s revisions you sent. Enclosed you will find our comments detailing the changes we have made. We hope that we have improved the paper in accordance with your recommendations. The
Sincerely yours,
Tatiana Vasilieva

Reviewer 2 Report
Comments and Suggestions for Authors
The comments are in the attached file.

Author Response
Dear Reviewer,
Thank you very much for the Referee’s revisions you sent. Enclosed you will find our comments detailing the changes we have made. We hope that we have improved the paper in accordance with the recommendations you gave.
Sincerely yours,
Tatiana Vasilieva

Reviewer 3 Report
Comments and Suggestions for Authors
The manuscript well presents the combination of EB and RF plasma treated polymer (PET), which shows greater hydrophilicity, resulting in a greater surface energy. This phenomenon is applied for the cell survival determination, which is novel and interesting. However, the manuscript has some discussion which is strange to me. The authors needs to elaborate my concerns as below before the manuscript is able to be accepted for publication.
1. Please clarify the flow rate of the oxygen into the chamber. Is there any Ar gas flow into the chamber to dilute the gas or help to ignite the plasma? Is there any bias power to direct the plasma bombardment?
2. For Fig 2, please calculate the surface roughness value.
3. From the results, I observe the HP-treated and RF-discharge treated samples have the similar results. Which sample would be beneficial for a hydrophilic and biocompatible surface?
4. Fig 6 seems confusing to me. For HP treated sample, why the cell survival rate increased over time? Also, please elaborate the fitting curve in Fig 6. Clearly, the green line does not go through the 72-hr point.
5. As the authors discussed, the treated samples shows similar results in cell survival rate, protein adhesion, and cell adhesion. In this case, why the combination of EB and RF treatment would be necessary. It seems that the treatment DOE is not necessary for the surface hydrophilicity change.
Author Response

(The authors gave the same response as above.)

Round 2
Reviewer 1 Report
Comments and Suggestions for Authors
The authors were able to address the comments. I recommend the publication of the manuscript.
Reviewer 3 Report
Comments and Suggestions for Authors
The authors have clarify my concern and the manuscript is in a good shape for publication